Subject Area:
bioinformatics/genomics/molecular biology

Keywords:
RNA secondary structure, natural selection, mRNA backfolding, PACKEIS

Author for correspondence:
David Rosenkranz
e-mail: rosenkranz@uni-mainz.de

# Widespread selection for extremely high and low levels of secondary structure in coding sequences across all domains of life

Daniel Gebert, Julia Jehn and David Rosenkranz

Institute of Organismic and Molecular Evolution iOME, Anthropology, Johannes Gutenberg University Mainz, Anselm-Franz-von-Bentzel-Weg 7, 55099 Mainz, Germany

DR, 0000-0001-5781-6201

Codon composition, GC content and local RNA secondary structures can have a profound effect on gene expression, and mutations affecting these parameters, even though they do not alter the protein sequence, are not neutral in terms of selection. Although evidence exists that, in some cases, selection favours more stable RNA secondary structures, we currently lack a concrete idea of how many genes are affected within a species, and whether this is a universal phenomenon in nature. We searched for signs of structural selection in a global manner, analysing a set of 1 million coding sequences from 73 species representing all domains of life, as well as viruses, by means of our newly developed software PACKEIS. We show that codon composition and amino acid identity are main determinants of RNA secondary structure. In addition, we show that the arrangement of synonymous codons within coding sequences is non-random, yielding extremely high, but also extremely low, RNA structuredness significantly more often than expected by chance. Taken together, we demonstrate that selection for high and low levels of secondary structure is a widespread phenomenon. Our results provide another line of evidence that synonymous mutations are less neutral than commonly thought, which is of importance for many evolutionary models.

## 1. Background

The genetic code of DNA uses units of three nucleotides (codons) to code for one amino acid. Since the number of possible codons exceeds the number of proteogenic amino acids, most amino acids are encoded not by a single codon but by several different codons. Therefore, mutations at the DNA level do not necessarily result in an altered amino acid sequence of the corresponding protein. These silent (synonymous) substitutions have long been assumed to be neutral in terms of natural selection [1]. However, silent substitutions will necessarily result in altered codon composition of a gene and further have the potential to alter a gene's GC content, both of which are features that can indeed be subject to selection [2]. Moreover, silent substitutions can change the secondary structure of an mRNA, thereby affecting the process of translation [3–6], and non-random patterns of secondary structures within protein coding genes in different species have been explained by natural selection [7–10].

However, the currently available data do not allow us to assess whether selection that acts on secondary structures within coding sequences represents a peculiarity of a few genomic loci in a limited number of species, or rather a widespread phenomenon affecting many genes in species throughout the domains of life. It is also not known whether selection acts only in one direction, favouring strong secondary structures as suggested by previous studies, or alternatively yields extremes at both ends of the spectrum.

To address these issues, we analysed protein coding sequences of 73 species representing all domains of life and further included more than 240 000 non-identical viral coding sequences. Using our newly developed software PACKEIS, we compared the predicted secondary structure of the evolutionary realized variants with that of corresponding artificial coding sequences that could have been realized in order to encode the same peptide sequence. We show that codon usage and amino acid identity both massively influence secondary structures. Beyond that, we identified protein coding sequences that exhibit extremely high or low structuring, compared with their artificial counterparts, independent of altered GC content or codon usage, which is due to a non-random arrangement of synonymous codons within the coding sequence. Importantly, these extreme solutions occur significantly more often than we would expect when assuming the absence of selection that favours structural extremes (structural selection). For the species under examination, we conservatively evaluate the fraction of protein coding sequences that are subject to structural selection to be, on average, 2–3%. We propose that altered structures of coding sequences affect a transcript's stability and/or its translation efficiency, which in turn are traits that evolution can act on, yielding the unexpectedly high number of structural extremes. A remarkably high number of coding sequences under structural selection was found in RNA viruses and we speculate that small RNA-based host immune systems have exerted a selective regime on viral genomes favouring highly backfolded transcripts to avoid targeting by anti-viral small RNAs [11,12].

## 2. Results

### 2.1. Approach and software development

Prior to our actual survey, we had to develop a software tool that allowed us to assess whether or not a realized open reading frame (original ORF, oORF) represents an extreme solution in terms of backfolding (base pairing with itself through self-complementarity), considering the alternative ORFs (aORFs) that could have been realized in order to encode the given peptide sequence based on the usage of synonymous codons. To this end, we have developed the highly parallelizable software PACKEIS, which compares the degree of backfolding (DBF) of the oORFs with that of a defined number of aORFs, yielding a DBF score ranging from 0 to 1; this score refers to the DBF in the light of alternative codon usage, with 0 representing extremely low structuredness and 1 representing extremely high structuredness. Details on the PACKEIS algorithm can be found in the Methods section.

### 2.2. ORFs exhibit extreme structures more often than expected by chance

Initially, we speculated that particularly ORFs of viral genes may exhibit high levels of backfolding in order to escape small RNA-based anti-viral responses of host immune systems. A piRNA-based immunity against viruses has been described in mosquito species such as *Aedes aegyptii*, and virus-derived siRNAs that confer anti-viral immunity can be found in plants and mosquitoes as well as in mice and human somatic cells [11–15]. We thus started our analyses with ORFs of polyproteins from 13 human-pathogenic mosquito-borne viruses [16]. Contrasting our expectation, we found that only the Edge Hill virus ORF exhibits DBF scores that imply a significant high DBF (DBF score$_{model0}$ = 0.99, DBF score$_{model2}$ = 1.00), while the ORFs of the other tested viral polyproteins have unremarkable DBF scores in the range of greater than 0.05 to less than 0.95 (figure 1*a*).

In order to assess whether ORFs with extremely high or low DBF scores occur significantly more often than expected by chance, we extended our analyses to a complete collection of non-identical viral ORFs ($n = 244\,314$) that are deposited at the GenBank sequence database managed by the National Center for Biotechnology Information (NCBI). Indeed, we observed a significant enrichment for ORFs with high DBF scores (figure 1*b*). While we would expect that, under neutral conditions, each DBF score from 0 to 1 in steps of 0.01 accounts for roughly the same number (1/101) of analysed ORFs, the fraction of ORFs with a DBF score of 1 accounts for 6.2%, 5.3% and 2.3% of all viral ORFs according to model0, model1 and model2, respectively (figure 1*b*). Despite the significant enrichment for ORFs with high DBF scores, the fact that the large majority of ORFs showed no signs of selection for strong secondary structures cast doubt on our initial speculation that small RNA-based immune responses have exerted strong selective regimes favouring highly backfolded ORFs.

Alternative explanations for favouring extreme levels of secondary structures are based on altered RNA stability and altered translation efficiency or a trade-off between both, which would be a fundamental principle whose footprints should be present in all organisms. We thus sampled 73 representatives from all domains of life and calculated DBF scores of their ORFs in a genome-wide manner (electronic supplementary material, table S1). We found that applying model2 yielded the most modest results and we will thus refer to the results that are based on model2 in the following, in order to give conservative estimates of the number of ORFs that we assume to be under structural selection and to exclude any effects related to GC content and biased codon usage. For the entirety of the analysed aORFs, the fraction of paired bases relative to the oORF follows a Gaussian distribution, ranging from 91% to 109% compared with the oORF. For 59 species, we observed a significant enrichment of ORFs with a DBF score$_{model2}$ > 0.95 (figure 1*c*). Of these, nine species additionally showed a significant enrichment of ORFs with a DBF score$_{model2}$ < 0.05. Nine species exhibited only a significant enrichment of ORFs with a DBF score$_{model2}$ < 0.05 but not ORFs with a DBF score$_{model2}$ > 0.95. For the remaining five species, significant enrichment for neither extremely high nor extremely low DBF scores could be observed (figure 1*c*; electronic supplementary material, table S1). When comparing the DBF scores of homologous genes across different species, we did not find any significant correlations, suggesting fluctuating structural selective forces on homologous genes along different phylogenetic branches (electronic supplementary material, table S2). In addition, we found no correlation between DBF scores$_{model2}$ and gene expression, neither at the transcript nor at the protein level (electronic supplementary material, table S3).

Remarkably, an enrichment for ORFs with DBF score$_{model2}$ < 0.05, suggesting selection that favours low levels of secondary structure, is present in 16 out of 33 sampled eukaryotic species (and seven out of eight plant

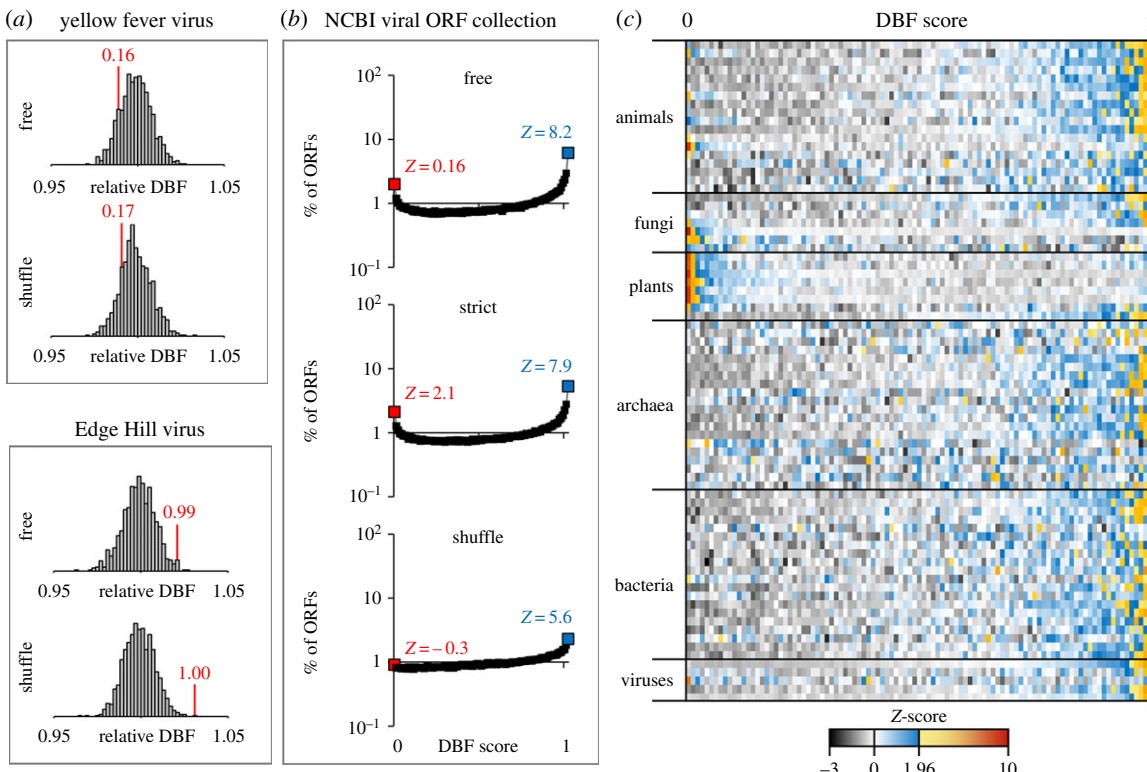

**Figure 1.** (*a*) DBF of aORFs from mosquito-borne RNA viruses for the purpose of illustration. The *X*-axis shows the distribution of DBFs relative to the average value of all aORFs. Genomes of the yellow fever virus and the Edge Hill virus both encode a single polyprotein. DBFs and the corresponding DBF scores are indicated in red. Only the ORF of the Edge Hill virus shows an exceptional DBF which is considerably higher than the corresponding aORFs (DBF score$_{model0}$ = 0.99, DBF score$_{model2}$ = 1.00). (*b*) The analysis of DBF scores for all available viral ORF sequences reveals a consistent and significant enrichment for extremely high DBF scores. (*c*) Lines in the heatmap represent species; rows represent DBF scores from 0 to 1 in steps of 0.01 using model2 (shuffle). The colour indicates row *Z*-scores with *Z*-scores above 1.96 ($p < 0.05$ for the two-tailed hypothesis) indicated in shades from yellow to red. Lines for viruses represent dsDNA, dsRNA, ssDNA, ssRNA($-$) and ssRNA($+$) viruses.

species), while the same applies to only one out of 20 sampled archaeal and one out of 20 sampled bacterial species. To quantify the fraction of ORFs that is presumably subject to structural selection within a given species, we summed up the fraction of ORFs with DBF scores$_{model2}$ < 0.05 and DBF scores$_{model2}$ > 0.95, taking only quantiles with *Z*-score $\geq$ 1.96 ($p < 0.05$ for the two-tailed hypothesis) into account. We then subtracted the share of ORFs expected to be allotted to the corresponding quantiles, assuming a uniform distribution of DBF scores in the absence of selection (see Methods). On average, we obtained similar fractions of ORFs under structural selection per species in the three domains of life ranging from 1.98% in archaea to 2.06% in eukaryotes and 2.54% in bacteria (figure 3*a*; electronic supplementary material, table S4).

Interestingly, the number of ORFs under structural selection from species representing all three domains of life is considerably lower than what we initially observed for virus ORFs, reviving the idea that small RNA-based immune systems may have contributed to the realized structuring patterns. If so, we would expect that particularly viruses that encode their genome in the form of single-stranded RNA, which represents a putative target for antisense small RNAs of a host, would be exposed to selective pressure that favours highly structured ORFs. We thus grouped viruses according to virus types into five classes (double-stranded (ds) DNA viruses, single-stranded (ss) DNA viruses, dsRNA viruses, ssRNA plus-strand viruses and ssRNA minus-strand viruses) and checked the structuring patterns of ORFs for each class separately (figure 2*a*).

Remarkably, the fraction of genes under structural selection is significantly higher in viruses and on average more than twice as high as in any of the three domains of life (figure 2*b*). Moreover, we found that ssDNA and dsDNA viruses show the lowest number of ORFs under structural selection within viruses, while ssRNA plus-strand and ssRNA minus-strand viruses exhibit the highest number of ORFs under structural selection. Notably, 11.5% of ORFs from ssRNA minus-strand viruses are presumably subject to structural selection, a value that surpasses that of any other of the 73 species under examination (figure 2*a*).

## 2.3. Codon composition and amino acid identity contribute to extreme secondary structures in ORFs

When we compared DBF scores obtained with different models, we were surprised by the observation that a considerable fraction of ORFs show extremely high or low score values applying model0 and model1 while showing only intermediate values when applying model2. Hence, we considered it unlikely that these ORFs were in fact subject to structural selection. Instead, we presumed that this pattern is caused by features other than ORF structure alone, features which must be differentially implemented in the different models.

In contrast with model2, model0 and model1 allow aORFs to be generated that differ in GC content and codon usage from the oORF. This applies in particular to oORFs that have an extremely biased codon usage compared with

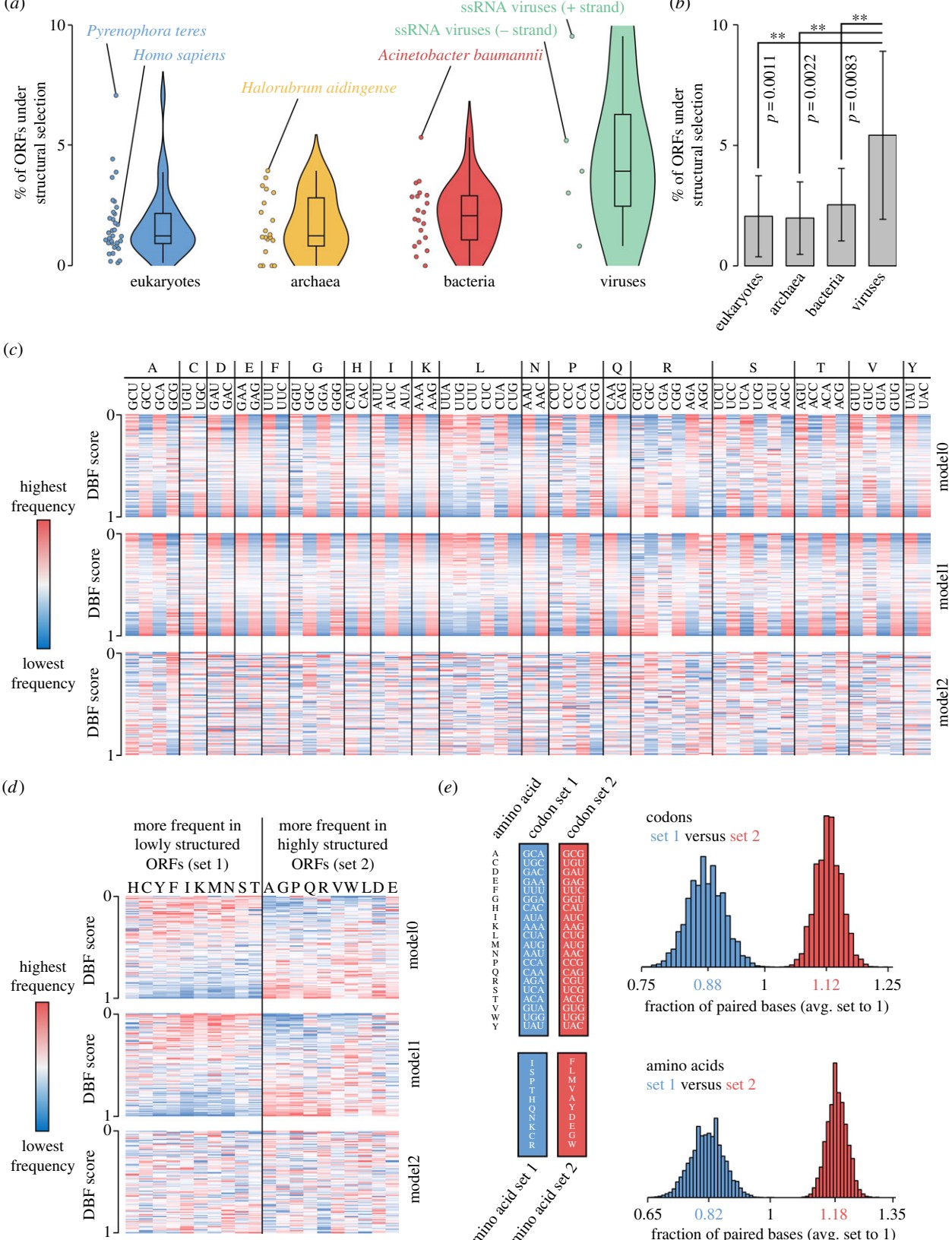

**Figure 2.** (*a*) Estimation on the fraction of ORFs under structural selection. (*b*) Virus ORFs are significantly more often under structural selection. *p*-values represent two-tailed *p*-values from unpaired *t*-tests. Error bars refer to standard deviation. (*c*) Codon frequencies in ORFs sorted by DBF score. Data exemplarily taken from *Mus musculus*. (*d*) Amino acid frequencies in ORFs sorted by DBF score. Data exemplarily taken from *M. musculus*. (*e*) Structuring of ORFs that code for identical peptide sequences but use different sets of codons (top) and structuring of ORFs that code for peptides being composed of different sets of amino acids (bottom).

a random codon usage (in the case of model0) or the global codon usage of the corresponding species (in the case of model1). We thus assumed that aberrant codon usage can lead to extreme structuring of ORFs and checked codon frequencies of oORFs separately for each DBF score quantile

and species, using the three different models. Remarkably, when applying model0 and model1, we observed that codons of all amino acids (except for M and W, which are encoded by only one codon) can be divided into those that are found more frequently in highly structured oORFs and

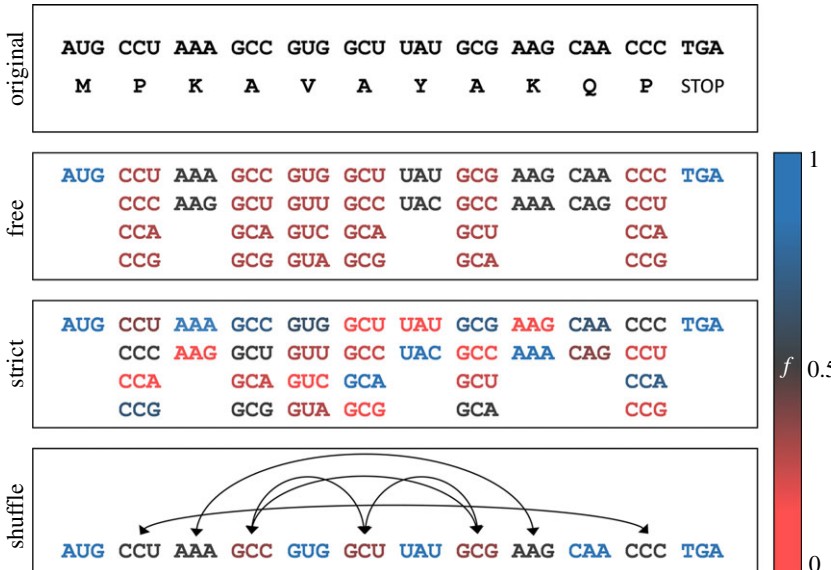

**Figure 3.** PACKEIS uses different models to generate artificial ORFs based on the oORF. Colours refer to the probability of a specific codon being placed at a given position. When applying model0 (free), PACKEIS uses equal probabilities for all codons of a specific amino acid. When applying model1 (strict), the probabilities are derived from the global codon usage of the species in question. When applying model2 (shuffle), codons of the oORF are randomly shuffled. Note that, in the above example, valine can only be encoded by GUG when applying model2 since no alternative valine codons are present in the given ORF.

those codons that are found more frequently in lowly structured oORFs, while a corresponding bias is absent when applying model2 (figure 2c). Interestingly, we made an analogous finding when focusing on different amino acids instead of codons, suggesting that amino acid identity also influences the degree of structuring (figure 2d). The above described division of codons for model0 and model1 invariably follows the combined number of G and C bases in the respective codons for each amino acid, where a higher codon GC content correlates with greater frequency in highly structured oORFs (electronic supplementary material, figure S2a), an observation recently also made by Fricke et al. [10]. Similarly, amino acids that are more frequently found in highly structured oORFs tend to exhibit higher mean GC shares in their respective codons ($p < 0.001$, Mann–Whitney U-test) (electronic supplementary material, figure S2b,c).

To verify the influence of divergent codon usage on secondary structure, we built a set of 10 000 random peptides with a length of 500 amino acids each. For each peptide, we constructed two corresponding aORFs, using only those codons that are most frequent in lowly structured oORFs (set 1) for the first aORF, and only those codons that are most frequent in highly structured oORFs (set 2) for the second aORF. Then we compared the DBF for both aORF groups as measured in the number of paired bases. Confirming our observation on codon bias across highly and lowly structured oORFs, we found that using different sets of synonymous codons can have a massive effect on the DBF with aORFs being composed of set 1 codons with an average number of paired bases corresponding to 0.88-fold of the total average (figure 2e). Accordingly, aORFs composed of set 2 codons have an average number of paired bases corresponding to 1.12-fold of the total average (figure 2e). Since species with large effective population size ($N_e$) are those where codon preferences correlate strongly with tRNA abundance [17], we checked for a connection between the number of genes under structural selection and $N_e$ but did not observe any correlation.

To check whether amino acid identity also contributes to oORF secondary structure, we built another two sets of 10 000 random peptides with a length of 500 amino acids each. Peptides in the first set were composed of only those 10 amino acids which are more frequently encoded in lowly structured oORFs (set 1 amino acids), while peptides in the second set were composed of only those 10 amino acids which are more frequently encoded in highly structured oORFs (set 2 amino acids). Equal probabilities for each codon of a given amino acid were used. As is the case for divergent codon usage, we found that amino acid identity is also an important determinant of oORF secondary structure, with aORFs of peptides being composed of set 1 amino acids with an average number of paired bases corresponding to 0.82-fold of the total average (figure 2e). Accordingly, ORFs of peptides composed of set 1 amino acids have an average number of paired bases corresponding to 1.28-fold of the total average (figure 2e).

## 3. Discussion

Owing to the degenerate nature of the genetic code, synonymous substitutions were initially regarded as neutral in terms of natural selection. Since then, a plethora of studies have demonstrated that synonymous codon usage can ultimately alter gene expression, clearly a trait that selection can act on.

As early as 1988, Shields et al. [18] found that the usage of synonymous codons among 91 Drosophila melanogaster genes varied greatly. Further, they observed that enhanced GC content due to preference for C-ending synonymous codons correlates with gene expression, a finding that was established previously also for different unicellular organisms [19–21]. Mechanistically, this can be explained by more stable and efficient transcription of GC-rich genes and the fact that unfavourable GC contents can trigger heterochromatization [22–24]. Apart from GC content, it is well

royalsocietypublishing.org/journal/rsob    Open Biol. **9**: 190020

known that codon usage also correlates with tRNA abundance, which is likely to be the outcome of a coevolution of codon usage and tRNA expression to optimize translation of highly expressed genes [25–28].

In contrast with our knowledge on how GC content and synonymous codon usage affects gene expression, far less is known about how secondary structure itself shapes gene expression patterns, which is possibly due to the fact that it is not trivial to disentangle these factors since one will influence the other and any difference in amino acid or synonymous codon usage will likely result in different degrees of RNA secondary structure [29]. In regard to this issue, we show here that ORFs with unusual levels of backfolding are indeed biased for specific sets of codons and amino acids, and our analysis of artificial coding sequences using these different codon and amino acid sets confirmed the close interweaving of codon usage bias, amino acid identity and extreme secondary structures. With these results in mind, we want to point out that any inferences on which traits evolution in fact acts on should be made with particular caution.

The possible role of mRNA secondary structure in the regulation of gene expression has been evaluated previously, though in a limited number of studies and species. Carlini et al. [30] compared two related drosophilid genes, Adh and Adhr, with respect to codon bias, expression and ability to form secondary structures. They noticed that the weakly expressed and weakly biased gene Adhr has a much stronger potential for backfolding than its heavily expressed and biased counterpart. Soon after, a more comprehensive study that compared the folding energies of original and corresponding artificial coding sequences generated by codon shuffling reported widespread selection for local RNA secondary structure, particularly in bacterial species but also in some archaeal and eukaryotic organisms [7]. Similar findings were subsequently presented for mammals [8]. Functional evidence for the importance of mRNA structures was provided by Kudla et al. [4], who showed that the stability of mRNA folding near the ribosomal binding site is a major determinant for the expression of a green fluorescent protein reporter encoded by a set of mRNAs that randomly differ at synonymous sites. A general correlation of folding energies and profiles of ribosomal density in Escherichia coli and Saccharomyces cerevisae emphasized the importance of mRNA secondary structure and translation efficiency [31]. Finally, a subtle large-scale analysis of coding sequences conducted by Fricke et al. [10] revealed that not only the amount of secondary structure but also its nature is non-random, with base pairing events between the first bases of two opposing codons being significantly underrepresented, suggesting the presence of selective forces.

Noteworthily, Hoede et al. [9] have proposed that selection also acts at the level of DNA structure during transcription and favours local intra-strand secondary structures to reduce the extent of transcriptional mutagenesis, a phenomenon that can be particularly observed at highly expressed genes. An important contribution to this complex issue was recently made by Lai et al. [32], who demonstrated that mRNAs as well as long non-coding RNAs intrinsically form secondary structures that result in short 5′- to 3′-end distances, which possibly affects the process of translation initiation.

Considering the available data, the unveiled widespread selection for high or low levels of secondary structure in coding sequences throughout the analysed species is not surprising, and we propose that evolutionary adjustment of the degree of secondary structure in ORFs contributes to fine-tuning of gene expression. If at all, one could argue that our estimates on the fraction of genes that are subject to this kind of selection appears surprisingly small. However, we want to emphasize that our estimates represent the lower limit, deduced from the enrichment of genes with extremely high or low DBF scores, additionally excluding those genes where we cannot rule out that selection acts on the level of codon usage and/or GC content instead of secondary structure alone. For these genes, we assume that restrictions on the amino acid sequence level prevent mRNAs from being optimally folded and that synonymous codon usage is exhaustively used to shift the mRNA towards the optimum. This would be a plausible explanation for the enrichment of genes at both ends of the DBF score spectrum. For an indeterminable fraction of genes, the optimal folding might be realized without requiring a suspicious arrangement of synonymous codons, though possibly not being less subject to structural selection that maintains the current state.

Interestingly, we found the highest number of genes under structural selection in RNA viruses. Many viral RNA structures, so-called cis-acting elements, are important for viral replication but are typically restricted to non-translated regions of the viral genome and the function of structures within coding sequences is poorly understood [33]. We speculate that highly structured viral coding sequences could be at least in part promoted by anti-viral host RNAi pathways. Many species have developed siRNA- and piRNA-based defence strategies to combat viral infections [11,12,34,35]. Since it has been shown that the target secondary structure is a major determinant of RNAi efficiency [36–38], we assume that the evolutionary arms race between viruses and hosts in many cases gives rise to viral transcripts that are characterized by a high DBF in order to provide as little attack surface as possible for single-stranded guiding RNAs.

In summary, our results demonstrate the close connection between codon usage bias, amino acid identity and RNA secondary structure. Moreover, using an as yet unrivalled broad data basis and an algorithm that excludes the effect of altered codon usage and GC content, we show that selection for extreme secondary structures within coding sequences is a widespread phenomenon throughout life and, with respect to viral ORFs, even beyond. Finally, with PACKEIS, we provide a tool that allows other researchers to easily conduct corresponding genome-wide analyses in any species of their choice not considered in the course of this study.

# 4. Methods

## 4.1. Data selection

In total, 73 representative species from the three domains of life (eukaryotes $n = 33$, archaea $n = 20$, bacteria $n = 20$) were selected, paying attention to a balanced representation of subclades within a given domain. ORF sequences from eukaryotes were downloaded from the Ensembl database (release 94, [39]). The longest transcripts for each gene were extracted using the custom Perl script select_longest_transcripts.pl. For archaea and bacteria species, we downloaded

royalsocietypublishing.org/journal/rsob Open Biol. 9: 190020

cDNA data and peptide data from Ensembl (release 94). We translated cDNA in all possible forward frames and checked for the presence of the resulting peptide sequences in the corresponding peptide dataset. Candidate ORFs with a match in the peptide dataset were collected for each species. The prediction of ORF sequences from cDNA and peptide datasets was conducted using the custom Perl script ORF_from_cDNA.pl.

Viral genome sequences were downloaded from NCBI GenBank. ORF sequences were extracted from GenBank files and converted into FASTA format using the custom Perl script GB_2_FASTA.pl. Viral sequences were further sorted into separate files according to viral classes (ssRNA positive strand, ssRNA negative strand, dsRNA, ssDNA, dsDNA) and hosts (algae, archaea, bacteria, environment, human, invertebrates, plants, protozoa, vertebrates) using the custom Perl script sort_viral_genomes.pl. All custom Perl scripts used in the course of this study are available at https://sourceforge.net/p/packeis.

Gene expression data were downloaded from the PaxDb Protein Abundance Database and EBI Expression Atlas [40,41].

## 4.2. The PACKEIS algorithm

In a first step, PACKEIS calculates the average probabilities for base pairing in predicted local RNA structures using algorithms of the ViennaRNA package [42]. Therefore, it runs RNAfold or RNAplfold (depending on the input ORF length) on a number of input sequences (FASTA file) and parses the resulting output files. Next, it calculates the DBF as measured in the fraction of paired bases within the oORF defined by the sum of the average base pairing probabilities for each position divided by the number of total bases. In a second step, PACKEIS generates a set of aORFs (default $n = 100$), each of which still codes for the same amino acid sequence, and calculates the DBF for each aORF as described above. By comparing the DBF of the oORF with those of the aORFs, PACKEIS outputs a measurand (DBF score) that allows us to assess the probability that the DBF of the oORF is a product of chance, where a DBF score of 0 means that none of the aORFs exhibits a lower DBF, while a DBF score of 1 means that none of the aORFs exhibits a higher DBF. The DBF score is calculated according to the following formula:

$$\text{DBF} - \text{score} = \frac{i/(i + j + k/2)}{j/(i + j + k/2)},$$

where $i$ refers to the number of aORFs with higher DBF and $j$ refers to the number of aORFs with lower DBF; $k$ refers to the number of aORFs with identical DBF compared with the oORF, so that the DBF score amounts to 0.5 in the case that all aORFs behave exactly as the oORF.

For the construction of aORFs, PACKEIS implements three different models. Model0 (free) uses equal frequencies for all synonymous codons of a specific amino acid. Model1 (strict) uses specified codon frequencies that reflect the codon usage of the species in question (or alternatively codon usage frequencies as defined by the user). Model2 (shuffle) constructs aORFs by shuffling those codons that are already present in a given oORF. In contrast with model0 and model1, codon frequencies and GC content are perfectly preserved in each of the aORFs compared with

the oORF when applying model2 (figure 3). Thus, applying model2 allows us to exclude the impact of aberrant codon usage or the GC content of a given oORF and to assess whether the present codons are arranged in a non-random fashion regarding the effect on secondary structure (figure 3).

PACKEIS produces a number of output files including a table that lists the DBF scores for each input sequence, a text file that shows the distribution of DBF scores from 0 to 1 in steps of 0.01, a table that refers to codon composition, amino acid identity and GC content for ORFs with a specific DBF score, and finally one text file per input sequence that lists the DBF scores for each of the corresponding aORFs.

To check whether the implemented models yield coherent results, we pairwise compared DBF scores for 27 628 *Arabidopsis thaliana* ORFs obtained when applying the three different models. Indeed, we observed a high degree of correlation across the results obtained by applying the different models as deduced from Pearson's correlation coefficients ranging from $r = 0.86$ to $r = 0.97$, supporting the general validity of the results (electronic supplementary material, figure S1).

The PACKEIS software including detailed documentation and test datasets is freely available at https://sourceforge.net/p/packeis and http://www.smallRNAgroup.uni-mainz.de/software.html.

## 4.3. DBF score calculation with PACKEIS

DBF scores were calculated using the PACKEIS software which was developed in the course of this study. PACKEIS will use RNAplfold to calculate base pairing probabilities based on local rather than global RNA folding, which is more reliable for larger sequences. Therefore, we set the minimum length [nt] of an input sequence for PACKEIS to run RNAplfold instead of RNAfold to 100 with the option -| 100. PACKEIS was run three times using different models for the construction of aORFs applying the option -m 0, -m 1 and -m 2, respectively.

## 4.4. Quantifying the number of ORFs under structural selection

We sectioned DBF scores into 101 quantiles ranging from 0 to 1 in steps of 0.01 ($Q_{0.00} \ldots Q_{1.00}$). ORFs that fell in the range 0–0.04 ($p < 0.05$) were considered as candidates for being subject to selection that favours low structuring. ORFs that fell in the range 0.96–1 ($p > 0.95$) were considered as candidates for being subject to selection that favours high structuring. The null hypothesis (absence of selection) was rejected if any of the lower or upper five quantiles showed a significant enrichment for ORFs. The summed fractions of genes in the lower ($s_{\text{low}}$) or upper ($s_{\text{high}}$) five quantiles with Z-scores $\geq 1.96$, deducting the share of ORFs that would be expected for each quantile assuming an even distribution in the absence of selection (1/101), was considered as the fraction of genes that is subject to structural selection ($S = s_{\text{low}} + s_{\text{high}}$) according to the following formula:

$$s_{\text{low/high}} = \sum_{k=i}^{j} f\left(z_{Q_{\frac{k}{100}}}\right) \cdot \left(a_{Q_{\frac{k}{100}}} - \frac{1}{101}\right),$$

where $f(Z < 1.96) = 0$ and $f(Z \geq 1.96) = 1$. For $s_{\text{low}}$ $i = 0$ and

royalsocietypublishing.org/journal/rsob    Open Biol. 9: 190020

$j = 4$; for $s_{high}$ $i = 96$ and $j = 100$. $a$ refers to the number of ORFs within the specific quantile. According to the definition of $f$, each of the lowest and highest five quantiles is taken into account only if ORFs from the corresponding quantile are significantly over-represented ($p < 0.05$, $Z \geq 1.96$). The term $z_{Q_{\frac{k}{100}}}$ refers to the Z-score of a specific quantile, e.g. $Q_{1.00}$ for $k = 100$. The function value $f\left(z_{Q_{\frac{k}{100}}}\right)$ becomes 0 for $z_{Q_{\frac{k}{100}}} < 1.96$. $s_{low}$ represents the number of genes where selection acts to reduce structuredness, while $s_{high}$ represents the number of genes where selection acts to enhance structuredness. Both fractions together represent the total number of genes $S$ that is subject to structural selection.

## 4.5. Generating random peptides and ORFs

To simulate the effect of using different sets of codons on secondary structure, we first sorted codons according to a bias in codon frequency across oORFs with high and low DBF scores$_{model1}$. A bias was attested in the case that we observed a steady increase or decrease in the average codon frequency from the upper five quantiles via the middle 91 quantiles to the lower five quantiles. Since the observed bias was not always consistent across all species under examination, we decided this based on the status in the majority of species (electronic supplementary material, table S5). Set 1 codons were those that were more frequent in oORFs with DBF scores$_{model1}$ ranging from 0 to 0.04 in the majority of species; set 2 codons were those that were more frequent in oORFs with DBF scores$_{model1}$ ranging from 0.96 to 1 in the majority of species. Amino acids were grouped into set 1 and set 2 amino acids accordingly.

To analyse the effect of using different sets of codons on secondary structure, we built a set of 10 000 random peptides with a length of 500 amino acids each. The first amino acid of each peptide was methionine. For each of the 10 000 random peptides, we constructed one corresponding aORF using set 1 codons and one corresponding aORF using set 2 codons. Then we predicted and compared the fraction of paired bases for aORFs composed of set 1 codons and aORFs composed of set 2 codons using the custom Perl script test_codon_sets.pl.

To analyse the effect of using different sets of amino acids on secondary structure, we built two sets of 10 000 random peptides with a length of 500 amino acids each, with the first peptide set being composed of set 1 amino acids and the second peptide set being composed of set 2 amino acids. One ORF for each of the 20 000 peptides was constructed with equal probabilities for possible codons of a given amino acid. We predicted and compared the fraction of paired bases for ORFs from set 1 peptides and ORFs from set 2 peptides using the custom Perl script test_aa_sets.pl. Note that, in contrast with the analysis of set 1 and set 2 codons, we did not compare sets of two aORFs that encode an identical peptide, but rather independent ORFs that code for different peptides (set 1 and set 2 peptides). All custom Perl scripts used in the course of this study are available at https://sourceforge.net/p/packeis.

Data accessibility. Figures and tables supporting this paper have been uploaded as electronic supplementary material.

Authors' contributions. D.G., J.J. and D.R. performed the relevant bioinformatics analyses and coded custom Perl scripts. D.G., J.J. and D.R. interpreted the results. D.G. and D.R. wrote the paper. The PACKEIS software was developed by D.R. D.R. is responsible for the study design and concept.

Competing interests. We declare we have no competing interests.

Funding. This work was supported by the intramural 'Impulsfonds' of the Johannes Gutenberg University, Mainz, Germany.

Acknowledgements. We thank Prof. Hans Zischler for helpful discussion and administrative support.

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
