## [Reviewer comments · Open Biology]

Review History

RSOB-19-0020.R0 (Original submission)

Review form: Reviewer 1

Recommendation

Accept with minor revision (please list in comments)

Are each of the following suitable for general readers?

- a) **Title**
Yes
- b) **Summary**
Yes
- c) **Introduction**
Yes

Is the length of the paper justified?

Yes

Should the paper be seen by a specialist statistical reviewer?

No

Is it clear how to make all supporting data available?

Yes

Is the supplementary material necessary; and if so is it adequate and clear?

Yes

Do you have any ethical concerns with this paper?

No

Comments to the Author

In their article, the authors investigate whether the coding sequence (CDS) of mRNAs is under evolutionary selection with respect to the RNAs structuredness, i.e. base pairing potential. For that purpose, the authors take over one million CDS from a total of 73 species from all domains of life (incl. viruses) and, for each of them, predict base pairing probabilities which in turn give rise to a single measure of structuredness, the degree of backfolding (DBF). To evaluate, whether a DBF for a particular CDS is significant, the authors compare it against alternative open reading frames (aORFs) obtained from three different background models. In particular, the aORFs consist of codons that preserve the encoded protein sequence, but are (i) selected from a uniform distribution, (ii) selected from the global codon usage of the respective organism, or (iii) derived from shuffling the original open reading frame (oORF) of the CDS. This comparison then gives rise to a DBF-score which ranges from 0 to 1 denoting that the oORF is less structured than background, or more structured than background, respectively.

From this analysis, the authors find that indeed many CDS seem to be significantly more (or less) structured than expected by the background model(s) throughout all kingdoms of life. In particular, viral CDS appear to be exceptionally structured. Furthermore, the authors identify two sets of codons that are over-represented in exceptionally structured, and unstructured ORFs. Similarly, two sets of amino acids are determined which, again, lead to structured and unstructured ORFs, respectively.

Along with their manuscript, the authors present a newly create software named **PACKEIS** that given a set of ORFs creates aORFs for each of the background models mentioned above. **PACKEIS** then computes the DBF and DBF-scores for the input data by utilizing the programs **RNAfold** or **RNAplfold** of the **ViennaRNA Package**.

The article is very well written and sound. However, I have some minor remarks that still need to be addressed.

1. The term 'high' and 'low' secondary structures that is used in the title and throughout the manuscript sounds rather odd to me. Especially in the title, it is unclear what high and low secondary structures could refer to. Since the authors associate structuredness in terms of the DBF with it, I would suggest adding the term 'structuredness' or something similar.

2. Page 3, Results, second paragraph:

Here, the definition of DBF appears in textual form. This should be moved to the Methods section, potentially associated with an equation.

3. Page 3, Results, second paragraph:

The definition of the DBF-score in this paragraph is not clear. The authors should add the formal definition to the Methods section. What confuses me is the fact that "...DBF-score of 0 means that none of the aORFs exhibits a lower DBF, while a DBF-score of 1 means that none of the aORFs exhibits a higher DBF". Assuming that a set of aORFs behaves exactly as the oORF, what would be the resulting DBF-score? Please clearly state the definition of the DBF-score!

4. Page 3, Results, paragraph 2 + 3:

Some of this text should rather be moved to the Methods section as it explains what the PACKEIS software actually does.

5. Page 3, Results, last paragraph:

The sentence "The PACKEIS software including ..." appears several times throughout the manuscript. While a little redundancy usually doesn't hurt, it might be better to state the 'Availability' of the software only once, either in the Methods section when you explain in detail what PACKEIS implements, or as a separate section 'Availability' at the end of the manuscript.

6. Page 9, Results:

for the amino acid identity analysis you create two sets of 10,000 random peptide sequences, one consisting of amino acids that appear in structured, and the other with amino acids that appear mostly in unstructured ORFs. From the text (even after reading the corresponding part in the Methods section), I don't fully understand what are the oORFs and aORFs in this analysis. After all, both are required to obtain the DBF-score if I'm not mistaken. Please clearly state what constitutes oORF and aORF.

7. Page 9, Results, last sentence before Discussion:

The Figure you want to reference here is most likely 3e, not Figure 2e.

8. Page 11, Methods section:

I'm missing a brief description what PACKEIS actually implements, and how it performs its tasks. What is the input, what is the output, what goes on in the black-box?

You mention RNAfold and RNAplfold which are used mutually exclusive depending on the length of the input ORF length. Please state whether these tools are run with default options or not. For RNAplfold you state that it is used with option '-l 100'. While this should probably read '-L 100' I wonder whether there are any other additional options passed to RNAplfold.

If not, I'd suggest adding the '-W' option with a window length at least 50nt larger than the value used for '-L' to level boundary effects for the computed base pair probabilities. Note here, that -L specifies the maximum allowed distance of two pairing nucleotides along the backbone, while the -W option specifies the window that is used to average the pairing probabilities. (see Bernhart et al. 2006, Bioinformatics).

9. Page 11, Methods section, "Quantifying the number of ORFs under structural selection"

The textual description of the s_{low}/s_{high} scores is a little bit confusing.

Please clarify:

(i) that you count the lowest/highest quantiles only if they are significant, i.e. $p < 0.05$ which translates to Z-score ≥ 1.96 .

(ii) what the s_{low}/s_{high} scores actually count. If I'm not mistaken, you are measuring the excess of ORFs in the five quantiles with respect to the expectation. Please rephrase this paragraph and clearly denote the parts of the text that correspond to the equation that follows this paragraph, i.e. state what $a_{\{Q k / 100\}}$ and $z_{\{Q k / 100\}}$ is.

10. Page 12, Methods section, "Generating random peptides and ORFs" As already mentioned for "6. Page 9, Results", the description for the second random peptide set is unclear. Please clarify what are the oORFs and corresponding aORFs that are required for DBF and DBF-score computation and averaging.

Review form: Reviewer 2

Recommendation

Accept with minor revision (please list in comments)

Are each of the following suitable for general readers?

- a) **Title**
Yes
- b) **Summary**
Yes
- c) **Introduction**
Yes

Is the length of the paper justified?

Yes

Should the paper be seen by a specialist statistical reviewer?

Yes

Is it clear how to make all supporting data available?

Yes

Is the supplementary material necessary; and if so is it adequate and clear?

Yes

Do you have any ethical concerns with this paper?

No

Comments to the Author

Gelbert et al present evidence that a small percent of protein coding sequences show a non-random degree of RNA secondary structure in species from all five kingdoms + viruses. The key method develops a metric for RNA structure based on the number of nucleotide pairs predicted by the ViennaFold algorithm and compares that to the number of nucleotide pairs predicted by three 100x-shuffled variants to estimate the random expectation. The best controlled shuffle variant maintains the same composition of the 61 codons, thus preserving GC content, amino acid content and synonymous codon usage. Other variants allow the effect of amino acid content and synonymous codon usage to be assessed. I find the principle claims to be true and interesting. The Discussion is well balanced, making it clear that it is challenging to determine if there is a selection on say amino acid content that is causing changes in structure or vice versa. The authors may well be correct that the true degree of secondary structure under selection in protein coding regions is larger than their analysis shows.

I hope the authors will consider the following.

1. Results, first sentence. The term “backfolding” is introduced without being defined. Why “back”, why not just folding. Please define the term when it is introduced.
2. I am not suggesting changing the DBF-score, but it does not distinguish between genes with many nucleotide pairs above or below random vs those with only one. The thresholding of greater than or less than the DBF of the oORF prevents this. Would it be useful to present in addition the distribution of, say, nucleotide pairs above or below the 100xmodel 2 background for, say, the genes DBF-score <0.05 or >0.95? Is there a Gaussian or, perhaps more likely, a single tailed distribution?
3. It might be worth pointing out that species with large effective population sizes are those where codon preferences correlate strongly with tRNA abundances, whereas those with higher mutational loads have codon usages that reflect neutral selection, such as GC content. e.g. dos Reis and Wernisch 2009 *mol biol evol* 26, 451. Would it be interesting to test if species with the structure bias correspond to those with higher or lower effective population sizes?
4. Given that synonymous codons for a given amino acid have closely related mono, di, and tri-nucleotide frequencies and that RNA di- and tri-nucleotides have differing base stacking and base pairing energies, it seems inevitable that any differences in amino acid or synonymous codon usage between genes would result in different degrees of RNA secondary structure. e.g. Mathews et al *J mol biol* 1999 288, 911. I would be tempted to make that explicit. (note to the editor, shuffle model 2 shows that some of the unexpected structure observed is not due to selection for amino acid or synonymous codon usage).
5. There is an interesting paper from Lai et al showing extensive folding of many mRNAs, including their protein coding region. *NATURE COMMUNICATIONS* | DOI: 10.1038/s41467-018-06792-z.
6. It is off putting that there is such a poor correlation in structure bias between orthologous genes. Codon usage, by contrast, is known to be well conserved among orthologs. This is one reason I am not compelled that the DFB-score is the most sensitive or best metric.

Decision letter (RSOB-19-0020.R0)

23-Apr-2019

Dear Dr Rosenkranz,

We are pleased to inform you that your manuscript RSOB-19-0020 entitled "Widespread selection for high and low secondary structure in coding sequences across all domains of life" has been accepted by the Editor for publication in *Open Biology*. The reviewer(s) have recommended publication, but also suggest some minor revisions to your manuscript. Therefore, we invite you to respond to the reviewer(s)' comments and revise your manuscript.

Please submit the revised version of your manuscript within 7 days. If you do not think you will be able to meet this date please let us know immediately and we can extend this deadline for you.

- 1) A text file of the manuscript (doc, txt, rtf or tex), including the references, tables (including captions) and figure captions. Please remove any tracked changes from the text before submission. PDF files are not an accepted format for the "Main Document".
- 2) A separate electronic file of each figure (tiff, EPS or print-quality PDF preferred). The format should be produced directly from original creation package, or original software format. Please note that PowerPoint files are not accepted.
- 3) Electronic supplementary material: this should be contained in a separate file from the main text and meet our ESM criteria (see <http://royalsocietypublishing.org/instructions-authors#question5>). All supplementary materials accompanying an accepted article will be treated as in their final form. They will be published alongside the paper on the journal website and posted on the online figshare repository. Files on figshare will be made available approximately one week before the accompanying article so that the supplementary material can be attributed a unique DOI.

Online supplementary material will also carry the title and description provided during submission, so please ensure these are accurate and informative. Note that the Royal Society will not edit or typeset supplementary material and it will be hosted as provided. Please ensure that the supplementary material includes the paper details (authors, title, journal name, article DOI). Your article DOI will be 10.1098/rsob.2016[last 4 digits of e.g. 10.1098/rsob.20160049].

- 4) A media summary: a short non-technical summary (up to 100 words) of the key findings/importance of your manuscript. Please try to write in simple English, avoid jargon, explain the importance of the topic, outline the main implications and describe why this topic is newsworthy.

Images

Data-Sharing

It is a condition of publication that data supporting your paper are made available. Data should be made available either in the electronic supplementary material or through an appropriate

repository. Details of how to access data should be included in your paper. Please see <http://royalsocietypublishing.org/site/authors/policy.xhtml#question6> for more details.

Data accessibility section

Sincerely,

The Open Biology Team
<mailto:openbiology@royalsociety.org>

Reviewer(s)' Comments to Author:

Referee: 1

Comments to the Author(s)

In their article, the authors investigate whether the coding sequence (CDS) of mRNAs is under evolutionary selection with respect to the RNAs structuredness, i.e. base pairing potential. For that purpose, the authors take over one million CDS from a total of 73 species from all domains of life (incl. viruses) and, for each of them, predict base pairing probabilities which in turn give rise to a single measure of structuredness, the degree of backfolding (DBF). To evaluate, whether a DBF for a particular CDS is significant, the authors compare it against alternative open reading frames (aORFs) obtained from three different background models. In particular, the aORFs consist of codons that preserve the encoded protein sequence, but are (i) selected from a uniform distribution, (ii) selected from the global codon usage of the respective organism, or (iii) derived from shuffling the original open reading frame (oORF) of the CDS. This comparison then gives rise to a DBF-score which ranges from 0 to 1 denoting that the oORF is less structured than background, or more structured than background, respectively.

From this analysis, the authors find that indeed many CDS seem to be significantly more (or less) structured than expected by the background model(s) throughout all kingdoms of life. In particular, viral CDS appear to be exceptionally structured. Furthermore, the authors identify two sets of codons that are over-represented in exceptionally structured, and unstructured ORFs. Similarly, two sets of amino acids are determined which, again, lead to structured and unstructured ORFs, respectively.

Along with their manuscript, the authors present a newly create software named PACKEIS that given a set of ORFs creates aORFs for each of the background models mentioned above. PACKEIS then computes the DBF and DBF-scores for the input data by utilizing the programs RNAfold or RNAplfold of the ViennaRNA Package.

The article is very well written and sound. However, I have some minor remarks that still need to be addressed.

1. The term 'high' and 'low' secondary structures that is used in the title and throughout the manuscript sounds rather odd to me. Especially in the title, it is unclear what high and low secondary structures could refer to. Since the authors associate structuredness in terms of the DBF with it, I would suggest adding the term 'structuredness' or something similar.

2. Page 3, Results, second paragraph:

Here, the definition of DBF appears in textual form. This should be moved to the Methods section, potentially associated with an equation.

3. Page 3, Results, second paragraph:

The definition of the DBF-score in this paragraph is not clear. The authors should add the formal definition to the Methods section. What confuses me is the fact that "...DBF-score of 0 means that none of the aORFs exhibits a lower DBF, while a DBF-score of 1 means that none of the aORFs exhibits a higher DBF". Assuming that a set of aORFs behaves exactly as the oORF, what would be the resulting DBF-score? Please clearly state the definition of the DBF-score!

4. Page 3, Results, paragraph 2 + 3:

Some of this text should rather be moved to the Methods section as it explains what the PACKEIS software actually does.

5. Page 3, Results, last paragraph:

The sentence "The PACKEIS software including ..." appears several times throughout the manuscript. While a little redundancy usually doesn't hurt, it might be better to state the 'Availability' of the software only once, either in the Methods section when you explain in detail what PACKEIS implements, or as a separate section 'Availability' at the end of the manuscript.

6. Page 9, Results:

for the amino acid identity analysis you create two sets of 10,000 random peptide sequences, one consisting of amino acids that appear in structured, and the other with amino acids that appear mostly in unstructured ORFs. From the text (even after reading the corresponding part in the Methods section), I don't fully understand what are the oORFs and aORFs in this analysis. After all, both are required to obtain the DBF-score if I'm not mistaken. Please clearly state what constitutes oORF and aORF.

7. Page 9, Results, last sentence before Discussion:

The Figure you want to reference here is most likely 3e, not Figure 2e.

8. Page 11, Methods section:

I'm missing a brief description what PACKEIS actually implements, and how it performs its tasks. What is the input, what is the output, what goes on in the black-box?

You mention RNAfold and RNApfold which are used mutually exclusive depending on the length of the input ORF length. Please state whether these tools are run with default options or not. For RNApfold you state that it is used with option '-l 100'. While this should probably read '-L 100' I wonder whether there are any other additional options passed to RNApfold.

If not, I'd suggest adding the '-W' option with a window length at least 50nt larger than the value used for '-L' to level boundary effects for the computed base pair probabilities. Note here, that -L specifies the maximum allowed distance of two pairing nucleotides along the backbone, while the -W option specifies the window that is used to average the pairing probabilities. (see Bernhart et al. 2006, Bioinformatics).

9. Page 11, Methods section, "Quantifying the number of ORFs under structural selection"

The textual description of the s_{low}/s_{high} scores is a little bit confusing.

Please clarify:

(i) that you count the lowest/highest quantiles only if they are significant, i.e. $p \leq 0.05$ which translates to Z-score ≥ 1.96 .

(ii) what the s_{low}/s_{high} scores actually count. If I'm not mistaken, you are measuring the excess of ORFs in the five quantiles with respect to the expectation. Please rephrase this paragraph and clearly denote the parts of the text that correspond to the equation that follows this paragraph, i.e. state what $a_{\{Q_k / 100\}}$ and $z_{\{Q_k / 100\}}$ is.

10. Page 12, Methods section, "Generating random peptides and ORFs" As already mentioned for "6. Page 9, Results", the description for the second random peptide set is unclear. Please clarify what are the oORFs and corresponding aORFs that are required for DBF and DBF-score computation and averaging.

Referee: 2

Comments to the Author(s)

Gelbert et al present evidence that a small percent of protein coding sequences show a non-random degree of RNA secondary structure in species from all five kingdoms + viruses. The key method develops a metric for RNA structure based on the number of nucleotide pairs predicted by the ViennaFold algorithm and compares that to the number of nucleotide pairs predicted by three 100x-shuffled variants to estimate the random expectation. The best controlled shuffle variant maintains the same composition of the 61 codons, thus preserving GC content, amino acid content and synonymous codon usage. Other variants allow the effect of amino acid content and synonymous codon usage to be assessed. I find the principle claims to be true and interesting. The Discussion is well balanced, making it clear that it is challenging to determine if there is a selection on say amino acid content that is causing changes in structure or vice versa. The authors may well be correct that the true degree of secondary structure under selection in protein coding regions is larger than their analysis shows.

I hope the authors will consider the following.

1. Results, first sentence. The term "backfolding" is introduced without being defined. Why "back", why not just folding. Please define the term when it is introduced.

2. I am not suggesting changing the DBF-score, but it does not distinguish between genes with many nucleotide pairs above or below random vs those with only one. The thresholding of greater than or less than the DBF of the oORF prevents this. Would it be useful to present in addition the distribution of, say, nucleotide pairs above or below the 100xmodel 2 background for, say, the genes DBF-score ≤ 0.05 or ≥ 0.95 ? Is there a Gaussian or, perhaps more likely, a single tailed distribution?

3. It might be worth pointing out that species with large effective population sizes are those where codon preferences correlate strongly with tRNA abundances, whereas those with higher mutational loads have codon usages that reflect neutral selection, such as GC content. e.g. dos Reis and Wernisch 2009 mol biol evol 26, 451. Would it be interesting to test if species with the structure bias correspond to those with higher or lower effective population sizes?

4. Given that synonymous codons for a given amino acid have closely related mono, di, and tri-nucleotide frequencies and that RNA di- and tri-nucleotides have differing base stacking and base pairing energies, it seems inevitable that any differences in amino acid or synonymous codon usage between genes would result in different degrees of RNA secondary structure. e.g. Mathews et al J mol biol 1999 288, 911. I would be tempted to make that explicit. (note to the editor, shuffle model 2 shows that some of the unexpected structure observed is not due to selection for amino acid or synonymous codon usage).

5. There is an interesting paper from Lai et al showing extensive folding of many mRNAs, including their protein coding region. NATURE COMMUNICATIONS | DOI: 10.1038/s41467-018-06792-z.

6. It is off putting that there is such a poor correlation in structure bias between orthologous genes. Codon usage, by contrast, is known to be well conserved among orthologs. This is one reason I am not compelled that the DFB-score is the most sensitive or best metric.

Author's Response to Decision Letter for (RSOB-19-0020.R0)

See Appendix A.

Decision letter (RSOB-19-0020.R1)

01-May-2019

Dear Dr Rosenkranz

We are pleased to inform you that your manuscript entitled "Widespread selection for extremely high and low levels of secondary structure in coding sequences across all domains of life" has been accepted by the Editor for publication in Open Biology.

Article processing charge

Please note that the article processing charge is immediately payable. A separate email will be sent out shortly to confirm the charge due. The preferred payment method is by credit card; however, other payment options are available.

Sincerely,

The Open Biology Team
mailto:openbiology@royalsociety.org

Appendix A

Dear Dr. Peter Parham,

Thank you for the positive feedback on our manuscript entitled “Widespread selection for high and low secondary structure in coding sequences across all domains of life”. We also want to thank the referees for their constructive criticism and valuable suggestions that helped us to improve the overall quality of the paper. We have revised the manuscript according to their suggestions. Please find our point-by-point response below.

Reviewer 1

1. We agree that the term high/low secondary structure is not optimal and now use the alternative terms “high/low structuredness” or “high/low levels of secondary structure” throughout the paper as suggested by the reviewer.
2. We have moved large parts of this section including the corresponding figure to the Results section and only provide a brief introduction including a description of the abbreviations oORF, aORF and DBF (Note that Fig. 1 is now Fig. 3, the numbering of the other figures was changed accordingly) We further agree with the reviewer’s opinion that readers would benefit from a clean mathematical description of the DBF-score and added an equation to the textual explanation.
3. We have mathematically clarified the definition of the DBF-score (see point 2.) and briefly elaborate on the case example mentioned by the reviewer since we believe that in fact many readers would come up with the same question.
4. We have moved paragraphs 2 and 3 to the Methods section.
5. We have removed redundant availability statements. Information on software availability can now be found in the Methods section, along with the description of the PACKEIS algorithm.
6. We regret this misunderstanding which was likely due to usage of the word aORF which is not appropriate for this section and was now replaced by ORF. We further added the following sentence to clarify this issue: “Note that in contrast to the analysis of set 1 and set 2 codons, we did not compare aORFs that encode an identical peptide, but rather independent ORFs that code for different peptides (set 1 and set 2 peptides)”.
7. This is true. We changed the text accordingly. Thanks for bringing this to our mind.
8. We thank the reviewer for this comment and have revised the according section to make the procedure more clearly for the readers. We explain input and output files of PACKEIS and further make clear that PACKEIS runs RNAfold and RNAplfold with default parameters. Unlike assumed by the reviewer, the option -l is passed to PACKEIS (not to RNAplfold) and states the maximum length of an input sequence to run RNAfold instead of RNAplfold. We have rephrased the corresponding explanation to exclude any misunderstandings.
9. We provide a more detailed explanation, elaborating on the two points raised by the reviewer, how we assess the number of genes under structural selection following the formula for calculation of s_{low} and s_{high} .
10. Based on the changes made according to point 6. we believe that we have now clarified this issue. We also made sure not to use the terms “DBF” or “DBF-score” but “fraction of paired bases”.

Reviewer 2

1. We agree that the term “folding” is more frequently used in the scientific literature compared to “backfolding”. Since we want to refer particularly to the base pairing of an mRNA with itself, rather than formation of loops or other 2-/3-dimensional structures, we prefer the term “backfolding” and added a definition of this term (“base pairing with itself through self-complementarity”) when using it for the first time as suggested by the reviewer.
2. We agree with the reviewer in so far as the DBF-score not allows to say something about how far a realized ORF structure is away from the ‘expectation’ as measured in additional/reduced paired bases. While this is an interesting question *per se*, we rather attempt to make a statement on

whether an oORF behaves significantly different compared to the group of aORFs. Our data show (we depicted some examples in figure 2a) that the fraction of paired bases for all aORFs of a given oORF fairly follows a Gaussian distribution with slightly differing variance across different oORFs. We can consider this Gaussian distribution as a probability density function and check the position of the oORF relative to this distribution, assuming 'unusual' folding if the oORFs falls into the upper or lower five percentiles of the distribution. This is independent of the absolute or relative distance from the center of the Gaussian distribution which would represent something like the expectation value for the number/fraction of paired bases. In other words, even a low number of additional paired bases compared to the aORFs average could be very unusual for one specific oORF, not or rarely reachable by random shuffling of synonymous codons (neutral evolution), while even a high number of additional paired bases could lay within the standard deviation of another oORF.

Nevertheless and in accordance with the reviewer's opinion, we think that information on how 'different' an oORF can be from the 'expectation' assuming neutral evolution is important and added the according information to Results section ("For the entirety of analyzed aORFs the fraction of paired bases relative to the oORF follows a Gaussian distribution ranging from 91% to 109% compared to the oORF").

3. This is indeed an interesting idea and we have again checked for any correlation regarding the estimated number of genes under structural selection and (putative) effective population size (N_e). However, the variance within unicellular eukaryotes or bacteria and archaea that typically have very large N_e is so high that it covers the range displayed by multicellular eukaryotes. Similarly, no difference between invertebrates (relatively large N_e) and vertebrates (relatively small N_e) can be observed. We added this information to the Results section.
4. We absolutely agree with this statement. We have aimed to impart this difficulty in the third paragraph of the results section but maybe missed to state it clear enough. We now spend some more words on this issue and cite the paper mentioned by the reviewer.
5. This is indeed an important paper. We have included the main findings in the discussion section and cite this study accordingly.

Other changes

1. Please note that we have corrected an error in the following line: "[...] where $f(Z < 1.96) = 0$ and $f(Z \geq 1.96) = 1$. For $s_{\text{low}} i = 0$ and $j = 4$, for $s_{\text{high}} i = 96$ and $j = 1$." where we have replaced " $j = 1$ " by " $j = 100$ ".
2. We realized that we have used the term "DBFscores" instead of "DBF-scores" several times in the Methods section. We now use the term "DBF-scores" throughout the manuscript.